# Modulation of Cellular Senescence in HEK293 and HepG2 Cells by Ultrafiltrates UPla and ULu Is Partly Mediated by Modulation of Mitochondrial Homeostasis under Oxidative Stress

**DOI:** 10.3390/ijms24076748

**Published:** 2023-04-04

**Authors:** Junxian Zhou, Kang Liu, Chris Bauer, Gerald Bendner, Heike Dietrich, Jakub Peter Slivka, Michael Wink, Michelle B. F. Wong, Mike K. S. Chan, Thomas Skutella

**Affiliations:** 1Institute for Anatomy and Cell Biology, Medical Faculty, Heidelberg University, 69120 Heidelberg, Germany; 2Department of Pharmacology of Chinese Materia Medica, China Pharmaceutical University, Nanjing 210009, China; 3MicroDiscovery, 10405 Berlin, Germany; 4Reviva s.r.o. Pekarska 6, 158 00 Prague, Czech Republic; 5Institute of Pharmacy and Molecular Biotechnology, Heidelberg University, 69120 Heidelberg, Germany; 6EW European Wellness International GmbH, 72184 Eutingen im Gäu, Germany

**Keywords:** cellular senescence, cell proliferation, SA-*β*-X-gal, senescence marker, intracellular ROS, mitochondrial ROS, mitochondrial fission

## Abstract

Protein probes, including ultrafiltrates from the placenta (UPla) and lung (ULu) of postnatal rabbits, were investigated in premature senescent HEK293 and HepG2 cells to explore whether they could modulate cellular senescence. Tris-Tricine–PAGE, gene ontology (GO), and LC–MS/MS analysis were applied to describe the characteristics of the ultrafiltrates. HEK293 and HepG2 cells (both under 25 passages) exposed to a sub-toxic concentration of hydrogen peroxide (H_2_O_2_, 300 μM) became senescent; UPla (10 μg/mL), ULu (10 μg/mL), as well as positive controls lipoic acid (10 μg/mL) and transferrin (10 μg/mL) were added along with H_2_O_2_ to the cells. Cell morphology; cellular proliferation; senescence-associated beta-galactosidase (SA-*β*-X-gal) activity; expression of senescence biomarkers including p16 INK4A (p16), p21 Waf1/Cip1 (p21), HMGB1, MMP-3, TNF-α, IL-6, lamin B1, and phospho-histone H2A.X (*γ*-H2AX); senescence-related gene expression; reactive oxygen species (ROS) levels; and mitochondrial fission were examined. Tris-Tricine–PAGE revealed prominent detectable bands between 10 and 100 kDa. LC–MS/MS identified 150–180 proteins and peptides in the protein probes, and GO analysis demonstrated a distinct enrichment of proteins associated with “extracellular space” and “proteasome core complex”. UPla and ULu modulated senescent cell morphology, improved cell proliferation, and decreased beta-galactosidase activity, intracellular and mitochondrial ROS production, and mitochondrial fission caused by H_2_O_2_. The results from this study demonstrated that UPla and Ulu, as well as lipoic acid and transferrin, could protect HEK293 and HepG2 cells from H_2_O_2_-induced oxidative damage via protecting mitochondrial homeostasis and thus have the potential to be explored in anti-aging therapies.

## 1. Introduction

The aging process can be studied in cell cultures that express typical characteristics of senescence [1,2]. In addition to intrinsic or replicative senescence caused by telomere erosion, various other conditions including oxidative stress, oncogenes, and tumor suppressors can induce extrinsic or premature senescence [1,3]. Several cytological, biochemical, and molecular changes accompany cellular senescence, such as growth arrest; cell morphology changes; activation of tumor suppressor networks; induction of senescence-associated beta-galactosidase activity; increased formation of reactive oxygen species (ROS) and senescence-associated heterochromatic foci (SAHF); secretion of various factors or senescence-associated secretory phenotype (SASP), such as pro-inflammatory cytokines (e.g., TNF-α and IL-6) and matrix metalloproteinases (MMPs) (e.g., MMP3); and increased autophagy [3,4,5]. The downregulation of nuclear lamina lamin B1 is another biomarker of induced senescence [5]. Two vital pathways, p53 and p16/Rb, participate in regulating the implementation of senescence-mediated growth arrest [6].

Mitochondria are dynamic structures in cells continuously undergoing fission and fusion mediated mainly by dynamin family proteins, which are essential for mitochondrial morphology and the maintenance of mitochondrial networks and functions [7,8]. Mitochondrial fission and fusion reflect the cell’s response to its physiological condition and are related to cellular quality control [9,10]. Mitochondrial oxidative phosphorylation produces ROS as a byproduct, which can cause damage to mitochondrial macromolecules, e.g., DNA, proteins, lipids, etc. [10,11,12]. ROS generated by mitochondria account for 90% of the total ROS in cells [12], and mitochondrial deficits caused by ROS contribute mainly to cellular senescence [10,11]. Mitochondria show decreased function with age [11,12], and more ROS can be generated by defective mitochondria [9]. In addition to excessive ROS production and imbalanced mitochondrial dynamics, multiple mitochondrial signaling pathways contribute to the perturbation of mitochondrial homeostasis and cellular senescence [4].

An important aim of medicine is to reverse aging or slow down the aging process. As anti-aging molecules, small molecules and protein therapeutics have been used for decades with advantages and limitations. Various methodologies are being developed to improve biotherapeutics [13,14,15]. The anti-aging activities of natural products and synthetic drugs have been widely studied in senescent cell cultures and in vivo [16,17,18]. Antioxidant enzymes and proteins possess anti-aging activities [19].

HepG2 cells reportedly become senescent by H_2_O_2_ induction and express a unique secretory phenotype, namely, downregulated Ki67 and increased p21, heterochromatin protein 1*β,* SA-*β*-X-gal, SASP, and total secreted protein levels [20]. In this study, we modified and optimized the method by using 300 μM of H_2_O_2_ to treat HEK293 and HepG2 cells for three days, followed by a four-day recovery culture to construct cellular senescence models. A pre-experiment was carried out in the constructed senescent cells with a series of protein probes, including ultrafiltrates from the thymus, cartilage, kidney, placenta, lung, and CNS of postnatal rabbits, to investigate whether they could modulate cellular senescence. The initial results indicated that ultrafiltrates from the placenta (UPla) and lung (ULu) had the best effects. Thus, Upla and Ulu, both with abundant proteins known for their antioxidant activities, were further studied in H_2_O_2_-induced senescent HEK293 and HepG2 cells. Several senescence characteristics were examined: cell morphology and proliferation, SA-*β*-X-gal, senescence markers, and senescence-related genes, intracellular ROS and mitochondrial ROS levels, and mitochondrial fission. Our data showed that both protein probes could modulate the senescence characteristics in HEK293 and HepG2 cells by protecting mitochondrial homeostasis under oxidative stress conditions.

## 2. Results

### 2.1. Proteomic Analysis of the 300 kDa Ultrafiltrates from the Placenta and Lung Tissues

The 300 kDa ultrafiltrates from the postnatal rabbit placenta and lung were isolated using a non-protein-denaturing extraction method. The concentrations of the ultrafiltrates were 1–5 μg/μL according to the Bradford assay. The results from Tris-Tricine–PAGE revealed that prominent detectable bands were between 10 and 100 kDa (Figure 1). Analysis by LC–MS/MS identified 150–180 proteins and peptides in the protein probes (Appendix A).

### 2.2. Gene Ontology (GO) Enrichment Indicated That UPla and ULu Consisted of Proteins Related to Oxidative Stress Responses and Glutathione Metabolism

GO analysis was performed on the proteins identified in the LC–MS/MS experiments to identify functional components using all three branches of the GO tree. To identify significantly enriched pathways, an over-representation analysis employing hypergeometric distribution was performed, as described by Blüthgen et al. [21]. GO analysis of the ultrafiltrates exhibited a distinct, general enrichment of proteins that were particularly associated with “extracellular space” and “proteasome core complex”. Furthermore, the GO terms “glutathione metabolic process”, “glutathione transferase activity”, “antioxidant activity”, and “removal of superoxide radicals” were observed (Figure 2, Appendix A).

### 2.3. Toxicity of the Protein Probes

The toxicity of UPla and ULu was investigated with the MTT assay, as previously described [22]. UPla and ULu were not toxic below 200 μg/mL in HEK293 cells and below 100 μg/mL in HepG2 cells after 72 h of incubation with the cells. The concentration used for both lipoic acid and transferrin (10 μg/mL) was not toxic for HEK293 and HepG2 cells). No DNA contamination of the ultrafiltrates was observed.

### 2.4. UPla and ULu Modulated H_2_O_2_-Induced Senescent Cell Morphology in HEK293 and HepG2 Cells

Treatment with 300 μM of H_2_O_2_ induced senescent cell morphology, namely causing enlarged flattened cells compared to the normal control cells (Figure 3). The positive controls lipoic acid and transferrin as well as the protein probes UPla and ULu modulated this morphology, causing a slight change.

### 2.5. UPla and ULu Reversed H_2_O_2_-Caused Cell Growth Arrest

The cell proliferation marker Ki67, which is abundant in all cell cycle phases (G1, S, G2, and mitosis) of proliferating cells but not in the G0 phase of quiescent cells [23], was used to investigate the proliferation of induced senescent HEK293 and HepG2 cells. As shown in Figure 4, H_2_O_2_ induction effectively reduced cell proliferation in both cell lines, suggesting cell cycle arrest. The addition of lipoic acid, Upla, and ULu reversed the cell proliferation decrease in senescent HEK293 cells; meanwhile, transferrin, UPla, and ULu modulated cell cycle arrest in senescent HepG2 cells.

### 2.6. UPla and ULu Decreased H_2_O_2_-Induced Senescence-Associated Beta-Galactosidase (SA-β-X-gal) Activity

After being exposed to H_2_O_2_, the cells became senescent, which was embodied by elevated SA-*β*-X-gal activity compared to that of the control group, as shown in Figure 5. Treatment with UPla, Ulu, as well as the positive controls reduced the number of H_2_O_2_-stimulated SA-*β*-X-gal-positive cells. The positive control Mito-TEMPO (20 μM), which is a mitochondrial-targeted antioxidant [24,25], had a similar effect.

### 2.7. UPla and ULu Suppressed Intracellular ROS Production Caused by H_2_O_2_

In this study, compared to the control group, H_2_O_2_ raised the level of oxidative stress, as evidenced by the higher intracellular ROS levels detected by 2′-7′-dichlorodihydrofluorescein diacetate (DCFH-DA) staining (Figure 6). DCFH-DA is widely used to measure intracellular ROS [26]. Treatment with UPla, Ulu, as well as lipoic acid, transferrin, and tauroursodeoxycholic acid (TUDCA) significantly reduced ROS overproduction. The anti-apoptotic molecule TUDCA, which reduces endoplasmic reticulum (ER) stress [27], was used as a positive control. It was reported that mitochondria–endoplasmic reticulum (ER) contacts play important roles in promoting cellular senescence and aging [28]. Our results suggested that ER stress might be involved in H_2_O_2_-induced oxidative damage.

### 2.8. UPla and ULu Reduced Mitochondrial ROS Generation Caused by H_2_O_2_

Mitochondrial ROS levels were elevated by H_2_O_2_ (Figure 7), which was indicated by strong red fluorescence due to mitochondrial superoxide. The mitochondria were universally marked with MitoTracker Green. Pre-treatment with UPla and ULu considerably lowered mitochondrial ROS levels, which was similar to the results obtained with lipoic acid, transferrin, and Mito-TEMPO.

### 2.9. UPla and ULu Modulated H_2_O_2_-Induced Cellular Senescence Markers

Cyclin-dependent kinase inhibitor 2A (p16) and cyclin-dependent kinase inhibitor 1 (p21), which slow down the cell cycle, are two key senescence markers [5]. p21 transcription is induced by p53 [6]. High mobility group box 1 (HMGB1) relocation from the nucleus to the cytoplasm and extracellular milieu is an early cellular response to senescence conditions, and this relocation facilitates SASP (e.g., IL-6 and MMP3 secretion) [29,30,31]. MMP3, a member of the matrix metalloproteinases (MMPs) that break down the extracellular matrix, is a SASP factor [5,32,33]. Inflammation is closely associated with aging. One of the most prominent SASP cytokines is interleukin-6 (IL-6) [33,34,35]. TNF-*α* is another SASP cytokine [36]. Nuclear lamin proteins build up the lamina that lines the inner surface of the nucleus. The loss of lamin B1, which is an essential lamin protein, is a biomarker of senescence and is dependent on the p53 or pRb pathway [37]. In the induction of cellular senescence, DNA damage accumulates and DNA double-stranded breaks induce *γ*-H2AX formation, which is a widely used senescence marker [38,39,40].

In our experiment (Figure 8), HEK293 cells exposed to H_2_O_2_ displayed significantly increased levels of p16, HMGB1, TNF-α, IL-6, and *γ*-H2AX, but only slightly increased levels of p21 and MMP3. Pre-treatment with ULu as well as with transferrin prevented the p16 increase. Lipoic acid and transferrin reversed the p21 increase, and UPla and transferrin suppressed HMGB1 acceleration. Lamin B1 protein expression was greatly lost in H_2_O_2_-induced senescent HEK293 cells and treatment with UPla, ULu, lipoic acid, and transferrin restored its expression. However, UPla and ULu did not seem to modulate the elevated levels of MMP3, TNF-α, Il-6, and *γ*-H2AX.

Similarly, H_2_O_2_-treated HepG2 cells (Figure 9) became senescent and showed an increase in p16, p21, HMGB1, MMP3, IL-6, and *γ*-H2AX expression. Pre-treatment with ULu and transferrin modulated the p16 increase. ULu and lipoic acid alleviated the rise in HMGB1 levels, whereas UPla and ULu as well as lipoic acid counteracted the MMP3 increase, and lipoic acid blunted the changes in IL-6 levels.

### 2.10. UPla and ULu Modulated Senescence-Related Gene Expression

The expression of the selected 37 senescence-related genes (Table 1) in HepG2 cells after 7 days of culture with H_2_O_2_ treatment, with and without UPla and ULu, was investigated by DNA-microarray analysis using the Fluidigm Biomark system. As shown in Figure 10, the expression of 25 genes was significantly altered by H_2_O_2_ induction and was significantly modulated by UPla and Ulu, with the only exceptions being TNF, SIRT1, VCAN, and MMP2 (*p* < 0.05; the significance was evaluated with GenEX software, but is not shown in Figure 10). Among these, the expression of P21, MMP3, PTEN, RB1, CDKN1B, NFκB1, MAPK14, AKT1, LONP1, SDHA, SDHC, PDHA1, ITPR2, TNF, TGFB1, IGF2, BMI1, VCAN, MMP2, DAO, PRODH, and SLC52A was upregulated and reversed by UPla and Ulu, with the only exceptions being TNF, VCAN, and MMP2 (Table 2). Meanwhile, there was a decline in the expression of CDKN2A/p16, SIRT1, which encodes the longevity protein SIRT1, and LMNB1 (Table 3).

The gene expression of CDKN1A/P21 and CDKN1B/P27 was upregulated by H_2_O_2_, whereas CDKN2A/P16 gene expression was downregulated; these results were similar to those of another study in which H_2_O_2_ induced cellular senescence [43]. These results combined with the overexpression of MAPK14, RB1, and PTEN suggested the involvement of the p53 and PTEN-p27Kip1 pathways in the cell cycle arrest in H_2_O_2_-induced HepG2 senescence [44,45]. Contrary to prediction, the expression of the mitochondria-related genes LONP1, SDHA, SDHC, PDHA1, and PRODH was elevated by H_2_O_2_, which could hardly be explained. DLP1 expression was not elevated significantly, but mitochondrial fission increased upon H_2_O_2_ treatment (Figure 11), which was probably mediated by other proteins. The overexpression of NFκB1, AKT1, ITPR2, and IGF2 indicated the involvement of multiple signaling pathways, such as inflammation, the insulin/IGF pathway, calcium homeostasis, endoplasmic reticulum (ER) stress, etc., in H_2_O_2_-induced HepG2 senescence.

### 2.11. UPla and ULu Ameliorated Mitochondrial Fission Increase Caused by H_2_O_2_

Mitochondria are dynamic organelles that undergo fission and fusion all the time to compensate for nonfunctional mitochondria and eliminate damaged organelles in response to metabolic changes [9,10]. In this study, H_2_O_2_ caused accelerated mitochondrial fission (Figure 11), which was reflected by increased numbers of spherical or fragmented mitochondria in HepG2 cells. Treatment with UPla, ULu, as well as with lipoic acid, transferrin, and Mito-TEMPO rescued cells from H_2_O_2_-induced oxidative stress.

## 3. Discussion

After H_2_O_2_ induction, HEK293 and HepG2 cells produced more ROS and developed senescent phenotypes. Both UPla and ULu modulated H_2_O_2_-induced changes in cell morphology, cell proliferation arrest, and X-gal staining. Some proteins and peptides have the potential to act as antioxidants that can inhibit lipid oxidation through different pathways, including the inactivation of ROS [46]. UPla and ULu both reduced the H_2_O_2_-mediated ROS elevation in HEK293 cells, reflecting the antioxidant activity of UPla and ULu that was associated with the GO terms “antioxidant activity” and ”removal of superoxide radicals.” According to LC–MS/MS analyses, superoxide dismutase (SOD) appeared in the two protein probes as a major component. SOD, a family of metalloenzymes, deactivates superoxide from mitochondria and other sources, turning it into O_2_ and H_2_O_2_ which are further decomposed to H_2_O by catalase or glutathione peroxidase [47,48]. In addition to Cu/ZnSOD and MnSOD in the intracellular space, extracellular Cu/ZnSOD (EC-SOD) exists in many species and defends the cells specifically against extracellular superoxide [47,49,50]. Research revealed the role of EC-SOD in controlling oxidative stress and suggested a more beneficial protective effect when EC-SOD was combined with other hydrogen-scavenging substances, such as catalase, glutathione peroxidase, etc. [51,52]. Moreover, cell metabolism is altered upon cellular senescence [53]. The GO terms “glutathione metabolic process” and “glutathione transferase activity” indicated the role of the two probes in modulating cell metabolism or acting as mitochondria-targeted antioxidants [54] in HEK293 and HepG2 senescence. *α*-lipoic acid, which can be rapidly absorbed by cells [25], has been reported to reduce ROS overproduction as a mitochondria-targeted antioxidant in various metabolic syndromes [54], which might explain its effects in this study.

Mitochondria are the main producers and targets of ROS [55]. It is reported that cellular senescence could be promoted by perturbation of mitochondrial homeostasis via six pathways: first, the excessive production of ROS; second, impaired mitochondrial dynamics; third, electron transport chain defects; fourth, bioenergetic imbalance/increased AMPK activity; fifth, decreased mitochondrial NAD+/altered metabolism; and sixth, mitochondrial Ca^2+^ accumulation. Electron transport chain defects contribute to ROS overproduction and decreased NAD+ levels, which further leads to a decrease in sirtuin activity. ROS overproduction and bioenergetic imbalance/increased AMPK activity could activate the p16/Rb and p53/p21 pathways [4]. In this study, UPla and ULu modulated accelerated mitochondrial fission, reversed H_2_O_2_-caused SIRT1 gene downregulation and ITPR2 gene increase which was associated with calcium influx. ULu reversed the p16 protein increase in both cells, and UPla and ULu modulated p21 gene expression in HepG2 cells.

In addition to modulating mitochondrial homeostasis under oxidative stress, UPla and ULu restored lamin B1 protein levels in HEK293 cells and LMNB1 gene expression in HepG2 cells. UPla suppressed HMGB1 protein accumulation in senescent HEK293 cells, and ULu modulated HMGB1 protein elevation in senescent HepG2 cells. UPla and ULu inhibited many H_2_O_2_-induced senescence-related gene expression changes. It has been reported that environmental stresses activate p38MAPK-dependent NF-κB signaling and cellular senescence. NF-κB signaling is the major signaling pathway that induces SASP. The TGF-*β-*TAK1 pathway activates NF-κB signaling, and NF-κB signaling and inflammatory mediators could be potentiated by HMGB1. Finally, p16INK4a is one of the inhibitors of NF-κB signaling [56]. The results in this study, namely H_2_O_2_-induced increases in the gene expression of MAPK14, NFκB1, MMP3, MMP2, TNF, and TGFB1 but not p16, as well as HMGB1 protein acceleration, are in accordance with these theories.

In conclusion, ultrafiltrates of the placenta and lung tissues from postnatal rabbits can modulate H_2_O_2_-induced cellular senescence in HEK293 and HepG2 cells through multiple and complex pathways, particularly by protecting mitochondrial homeostasis under oxidative stress conditions. Further studies could be performed in vivo to investigate their anti-senescence and anti-aging functions.

## 4. Materials and Methods

### 4.1. Materials and Cells

Hydrogen peroxide, 2′,7′-dichlorfluorescein-diacetat (DCFH-DA), Mito-TEMPO, 3-(4,5-dimethylthiazol-2-yl)-2,5-diphenyltetrazolium bromide (MTT), lipoic acid, and transferrin were obtained from Sigma-Aldrich (Darmstadt, Germany). TUDCA was bought from Avanti Polar Lipids, Inc. (Alabaster, AL, USA). MitoTracker™ Red CMXRos; MitoSOX™ Red; DAPI; Goat anti-Mouse IgG (H + L), Superclonal™ Recombinant Secondary Antibody, Alexa Fluor™ 488; and goat anti-Rabbit IgG (H + L), Cross-Adsorbed Secondary Antibody, Alexa Fluor™ 488 were purchased from Thermo Fisher Scientific (Karlsruhe, Germany). MitoTracker^®^ Green FM was obtained from Cell Signaling Technology (Leiden, The Netherlands). Mouse anti-Ki-67 antibody was bought from BD Sciences (BD Pharmingen™, Heidelberg, Germany).

Human embryonic kidney (HEK) 293 cells were kindly provided by Dr. Rolf Sprengel (Department of Molecular Neurobiology, Max Planck Institute for Medical Research, Heidelberg, Germany). Human hepatocellular carcinoma cell line HepG2 was a kind gift from Prof. Stephan Urban (Department of Infectious Diseases, Molecular Virology, Centre for Integrative Infectious Disease Research (CIID), Heidelberg University, Heidelberg, Germany).

### 4.2. Production of UPla and ULu

Frozen (no more than −20 °C) 1–2-day-old specimens of laboratory-grade postnatal rabbits (*Oryctolagus cuniculus*, broiler rabbit lineage M91) with their placentas were received, thawed, washed, and dried off. The rabbits were maintained for use in broiler rabbit production systems by interline hybridization from New Zealand White rabbits and were bred at the Institute of Small Farm Animals at the Research Institute of Animal Production Nitra, Slovakia. The placenta (for UPla) and lungs (for ULu) were removed and placed into sterile test tubes with saline solution and homogenized for 5 min. The homogenized solutions were transferred into clean test tubes (50 mL) and centrifuged at 6000 rpm at 10 °C for 10 min. After pre-filtration with ceramic filters, micro- and ultrafiltration were performed. Namely, using a benchtop crossflow system (Sartorius Sartoflow Benchtop system, Göttingen, Germany), the samples were micro-filtered through a 0.45 kDa cassette followed by ultrafiltration with a 300 kDa cut-off cassette. The final permeates, UPla and ULu, were separately collected into sterile flasks for use. The whole manufacturing process was standardized and performed under cooling conditions.

### 4.3. Bradford Assay

A 10 µL sample was pipetted in triplicate into the wells of a microplate with standard bovine serum albumin (BSA) protein solutions (50, 100, and 150 g/mL) or Milli Q water. Then, 100 µL of Bradford reagent (Proteome Factory PS-2009) [57] was added to each sample and mixed well. After 15 min, the absorbance was assessed at 595 nm using a spectrophotometer (Titertek Multiscan TCC/340, Thermo Fisher Scientific, Langenselbold, Germany). A standard curve constructed in Excel by plotting the absorbance values (*y*-axis) versus their concentrations in µg/mL (*x*-axis) was used to calculate the sample concentrations.

### 4.4. Tris-Tricine–PAGE

Tricine–SDS-PAGE is commonly used to separate proteins in the mass range of 1–100 kDa. It is the preferred electrophoretic system for proteins with a resolution smaller than 30 kDa. A sample (5 µg or 12 µL) was mixed with 4× Laemmli sample buffer (250 mM Tris, pH 6.8; 12% glycerol; 4% sodium dodecyl sulfate (SDS); 10% beta-mercaptoethanol; 0.05% bromphenol blue) and then boiled for 5 min at 95 °C. Following the method described by Schägger and Jagow [58], samples were separated on a gel after cooling (7 × 8 cm; 4% stacking gel [4% acrylamide: bisacrylamide (29:1)]; 68 mM tris, pH 6.8; 0.2% SDS; 0.2% N,N,N′,N′-tetramethylethylenediamine (TMED); 0.03% ammonium persulphate (APS) with 18% separating gel [18% acrylamide: bisacrylamide (32:1)]; 1 M Tris, pH 8.45; 0.1% SDS; 14% glycerine; 0.05% TMED; 0.05% APS) at 150 V for 180 min (Mini-protein II Dual Slab Cell, Bio-Rad, Feldkirchen, Germany). The protein sizes were calculated by comparing the migration of the protein bands to molecular mass standards (Mark12, Thermo Darmstadt, Germany) after the gel was stained with mass spectrometry-compatible silver staining (Proteome Factory PS-2001, Proteome Factory AG, Berlin, Germany). The gel was scanned using a ScanMaker 9800 XL plus scanner (Microtek International Inc., Hsinchu, Taiwan) with a transparency adaptor at a resolution of 150 dpi.

### 4.5. Liquid Chromatography–Mass Spectrometry/Mass Spectrometry (LC–MS/MS) Analysis

LC–MS/MS analysis of the proteins was performed using an UltiMate 3000 nano HPLC system (Thermo Scientific, Germering, Germany). Water served as solvent A and acetonitrile served as solvent B (with 0.1% formic acid additions). Aliquots of 200 µg proteolytic peptides obtained via trypsin were loaded onto a trapping column (maintained at 35 °C) for desalting (Thermo Scientific, Dionex, PepMap C18, Sunnyvale, CA, USA) before the peptides were separated by gradient elution on a 500 × 0.075 mm column (0.5 µL/min, held at 50 °C; Reprosil C18-AQ, Dr. Maisch) from 12% to 40% B. The column effluent was directed to an Orbitrap Velos mass spectrometer via a nanoelectrospray ion source (Thermo Scientific, Bremen, Germany). Up to 10 MS/MS spectra from ions of interest (charge states +2 and higher) were data-dependently captured in the instrument’s linear ion trap, while survey scans were detected at a notional resolution of R = 60,000. The total acquisition time of each analysis was 45 min.

### 4.6. Database Search with Mascot

Proteins were identified using database searches against *Oryctolagus cuniculus* sequences from the NCBI protein database (National Center for Biotechnology Information, Bethesda, MD, USA) or the UniProt proteome UP000001811 [59] using an MS/MS ion search of the Mascot search engine (version 2.6.2, Matrix Science, London, UK). Only peptides that matched with a score of 20 or above were accepted.

### 4.7. Bioinformatics

To map the proteins to GO terms, the R packages “biomaRt” (version 2.38.0) [60] and “UniProt.ws” (version 2.22.0) [61] were used. We detected significantly enriched GO terms using an over-representation approach as implemented in the R package “topGO” (version 2.32.0) [62]. The “topGO” package performs enrichment analysis of Gene Ontology terms while accounting for the topology of the GO graph. We used *p*-values corrected for multiple testing. To assign proteins to KEGG pathways, we used the R packages “KEGG.db” (version 3.2.3) [63] and “KEGGREST” (version 1.20.1) [64]. To identify significantly enriched pathways, we performed an over-representation analysis employing hypergeometric distribution, as described by Blüthgen et al. [21].

### 4.8. Cell Culture and Cell Viability with MTT Assay

HEK293 cells were cultured in Dulbecco’s Modified Eagle Medium (Gibco™, Thermo Fisher Scientific, Karlsruhe, Germany) supplemented with 10% fetal bovine serum (FBS, Capricorn Scientific, Ebsdorfergrund, Germany) and 100 U/mL of penicillin and streptomycin (Gibco™, Thermo Fisher Scientific, Karlsruhe, Germany). HepG2 cells were cultured in Eagle’s Minimum Essential Medium (LGC, Wesel, Germany) supplemented with 10% FBS and 100 U/ML of penicillin and streptomycin.

The toxicity of H_2_O_2_ and the single modulators lipoic acid, transferrin, UPla, and ULu was tested in HEK293 and HepG2 cells. In total, 3 × 10^4^ cells were seeded in 96-well plates and incubated at 37 °C for 24 h. Hydrogen peroxide with or without the single substances was added, and the cells were further incubated at 37 °C for 72 h. Afterwards, the media were removed, and 0.5% MTT prepared in media was added to each well. After 2–4 h of incubation at 37 °C, the media were replaced with DMSO, and the absorption was read at 570 nm using the Tecan NanoQuant Infinite M200 PRO Plate Reader (Tecan, Männedorf, Switzerland).

### 4.9. Cellular Senescence Induction and Cell Morphology Observation

The sub-toxic concentration of H_2_O_2_ (300 μM) was used to treat normal HEK293 and HepG2 cells (both under 25 passages) for three days with media refreshed every day, followed by a four-day recovery culture in regular media without H_2_O_2_, whereby the media were changed one day before the assay. Thus, HEK293 and HepG2 cells were regarded as being induced to become premature senescent cells. The substances UPla (10 μg/mL), ULu (10 μg/mL), as well as lipoic acid (10 μg/mL), and transferrin (10 μg/mL) were added along with H_2_O_2_ to the cells. Lipoic acid was used as a positive control. The glycoprotein transferrin, which specifically binds ferric ions [46], was chosen as a positive protein control. In all the experiments, the same concentration (10 μg/mL) was used for the single substances. Cell morphology was studied under light microscopy (Zeiss AxioVert.A1, Oberkochen, Germany).

### 4.10. SA-β-X-Gal Staining

The senescence-associated beta-galactosidase activity in the induced senescent HEK293 and HepG2 cells, with or without the single substances, was measured using the Senescence Cells Histochemical Staining Kit (Sigma-Aldrich, Darmstadt, Germany), following the kit’s instructions. Cells were analyzed via bright-field microscopy (Zeiss Axiovert 135 TV, Oberkochen, Germany) and photos were taken. The percentage of SA-*β*-X-gal-positive cells was calculated to represent the results.

### 4.11. Detection of Cellular Proliferation with Ki67 Antibody

HEK293 and HepG2 cells pre-treated with the single substances lipoic acid (10 μg/mL), transferrin (10 μg/mL), UPla (10 μg/mL), and ULu (10 μg/mL) plus H_2_O_2_ (300 μM) were fixed with 4% paraformaldehyde (PFA) for 20 min and then treated with 0.1% triton for 15 min. After washing the cells with phosphate-buffered saline (PBS) three times, 10% goat serum was added to the cells, and the cells were kept at room temperature for 1 h. Then, purified mouse anti-Ki-67 antibody (BD Pharmingen™, Heidelberg, Germany) prepared in PBS was added to the cells, and the cells were incubated at 4 °C overnight. After washing the cells with PBS, Goat anti-Mouse IgG (H + L), Superclonal™ Recombinant Secondary Antibody, Alexa Fluor™ 488 (Invitrogen, Thermo Fisher, Karlsruhe, Germany) was added to the cells for further incubation at room temperature for 1 h. DAPI was used to stain the nuclei for 5 min. Then, the cells were studied under confocal microscopy (Zeiss LSM700, Oberkochen, Germany).

### 4.12. Intracellular ROS

HEK293 cells pre-treated with lipoic acid (10 μg/mL), transferrin (10 μg/mL), TUDCA (100 μM), UPla (10 μg/mL), and ULu (10 μg/mL) plus H_2_O_2_ were treated with 25 μM of DCFH-DA for 30 min. After washing with PBS twice, the accumulation of DCF was visualized at 488 nm under fluorescence microscopy (Zeiss Axiovert 200, Oberkochen, Germany).

### 4.13. Mitochondrial ROS

HEK293 cells pre-treated with lipoic acid (10 μg/mL), transferrin (10 μg/mL), Mito-TEMPO (20 μM), UPla (10 μg/mL), and ULu (10 μg/mL) with H_2_O_2_ were rinsed with PBS and cultured in pre-warmed DMEM medium containing 30 nM of MitoTracker Green and 4 μM of MitoSOX™ Red at 37 °C for 30 min. After washing with PBS, fresh DMEM was added and the cells were observed under confocal microscopy (Zeiss LSM700, Oberkochen, Germany). The mean values of the integrated density of red fluorescence due to MitoSOX™ Red oxidization by mitochondrial superoxide were evaluated.

### 4.14. Immunofluorescence Staining and Confocal Microscopy

HEK293 and HepG2 cells pre-treated with single substances lipoic acid (10 μg/mL), transferrin (10 μg/mL), UPla (10 μg/mL), and ULu (10 μg/mL) plus H_2_O_2_ (300 μM) in 24-well plates were fixed with 4% PFA for 20 min and then treated with 0.1% triton for 15 min. After washing the cells with PBS three times, 10% goat serum was added. After 1 h of incubation, the senescence marker antibodies p16 INK4A (D3W8G) Rabbit mAb, p21 Waf1/Cip1 (12D1) Rabbit mAb, phospho-histone H2A.X (Ser139), lamin B1 (D9V6H) Rabbit mAb, HMGB1 (D3E5) Rabbit mAb, IL-6 (D3K2N) Rabbit mAb, TNF-*α* (D5G9) Rabbit mAb, and MMP-3 (D7F5B) Rabbit mAb from the Senescence Marker Antibody Sampler Kit (Cell signaling, Leiden, The Netherlands) were added, and the cells were incubated at 4 °C overnight. After washing with PBS, Goat anti-Rabbit IgG (H + L), Cross-Adsorbed Secondary Antibody, Alexa Fluor™ 488 (Invitrogen, Thermo Fisher, Karlsruhe, Germany) was added, and the cells were incubated for 1 h at room temperature. DAPI was used to stain nuclei for 5 min. The images were recorded with a confocal microscope (Zeiss LSM700, Oberkochen, Germany).

### 4.15. Gene Expression Analysis by Microfluidic High-Throughput Real-Time PCR

HepG2 cells pre-treated with H_2_O_2_ with or without UPla and ULu were harvested by trypsinization. Following the kit’s instruction, total RNA was extracted from the cell pellets using a NucleoSpin RNA Mini kit for RNA purification (MACHEREY-NAGEL, Dueren, Germany). The purity of the RNA was verified using a NanoDrop (Thermo Fisher Scientific, Karlsruhe, Germany). The integrity of the RNA samples was verified using the Agilent TapeStation system (Agilent Technologies, Santa Clara, CA, USA). The samples were diluted to a final concentration of 20 ng/µL. For reverse transcription and pre-amplification, the inventoried TaqMan assays (Thermo Fisher Scientific, Karlsruhe, Germany) were pooled to a final concentration of 0.2× for each of the assays in TE buffer with a pH of 7. On a 96-well plate, a reaction mix was prepared with 5.0 µL of RXN direct buffer 2×, 2.5 µL of assay pool 0.2×, 0.2 µL of Superscript III Platinum Taq mix (Invitrogen, Thermo Fisher Scientific, Karlsruhe, Germany), 1.3 µL of RNase free water, and 1 µL (20 ng) of total RNA. Sequence-specific reverse transcription was performed at 50 °C for 15 min. The reverse transcriptase was inactivated by heating it to 95 °C for 2 min. In the same tube, cDNA underwent limited sequence-specific amplification by denaturing at 95 °C for 15 sec and 18-cycle annealing and amplifying at 60 °C for 4 min. These pre-amplified products were 5-fold diluted in water before analysis with Universal PCR Master Mix (Thermo Fisher Scientific, Karlsruhe, Germany) and inventoried TaqMan gene expression assays using a 96.96 Dynamic Array Microfluidic Chip on a Biomark HD system (Fluidigm/Standard BioTools, San Francisco, CA, USA). Each sample was analyzed in three technical replicates. GenEx statistical analysis was performed as previously described [65]. 

### 4.16. Mitochondrial Fission

HepG2 cells pre-treated with lipoic acid, transferrin, Mito-TEMPO, Upla, and ULu were treated with 200 nmol/mL of MitoTracker™ Red CMXRos for 30 min and visualized using the confocal microscope (Zeiss LSM700, Oberkochen, Germany). Ninety cells were counted, and the percentage of cells undergoing mitochondrial fission was compared among groups. Spherical or fragmented mitochondria were considered to be undergoing mitochondrial fission in contrast to normal mitochondria presenting the reticular form [66].

### 4.17. Statistical Analysis

All of the results are expressed as the mean ± SD from three independent experiments. The quantification of the fluorescence densities indicating intracellular ROS was performed using ImageJ 152-win-java8 (NIH, Bethesda, MD, USA) and GraphPad Prism 7 (GraphPad Software, San Diego, CA, USA). Ct values obtained from the Biomark system were analyzed using GenEx software (MultiD). Statistical significance was assessed with one-way ANOVA analysis. The difference was regarded as significant when *p* < 0.05.

## Figures and Tables

**Figure 1 ijms-24-06748-f001:**
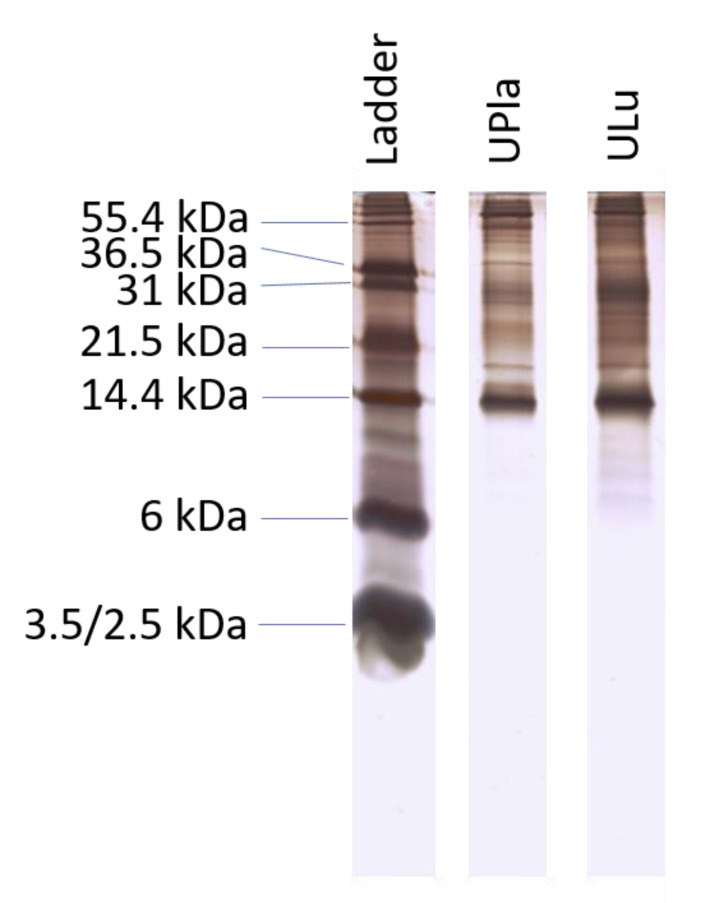
Ultrafiltrates from placenta and lung samples were analyzed by Tris-Tricine–PAGE with a nominal 1 µg per lane. The result is representative of similar results of three independent experiments.

**Figure 2 ijms-24-06748-f002:**
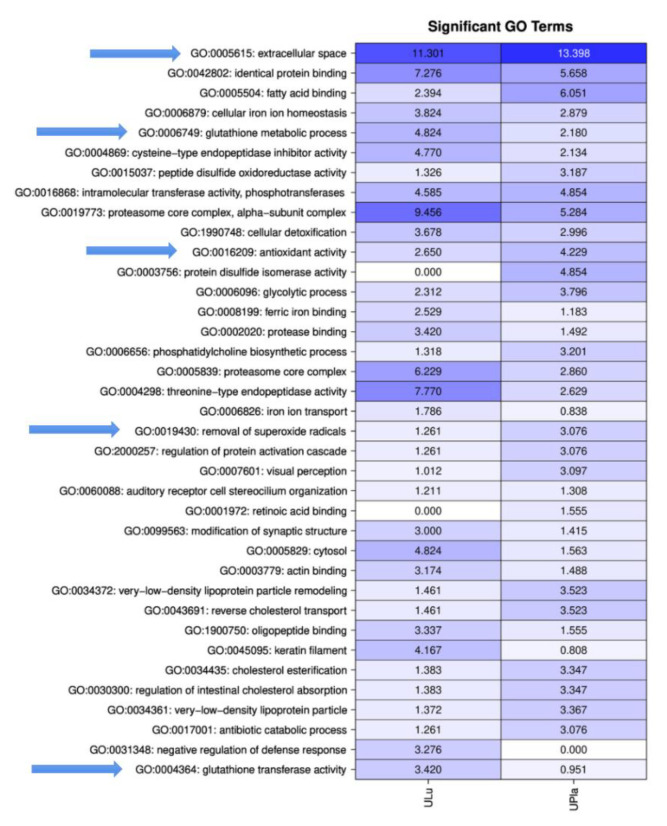
Matrix with −log_10_ *p*-values of significantly identified GO terms for ULu and UPla. Columns indicate the different tissues; rows indicate the GO terms. All GO terms with a *p*-value < 0.001 were used. Values > 1.3 are significant. Arrows mark important GO terms related to oxidative stress and senescence.

**Figure 3 ijms-24-06748-f003:**
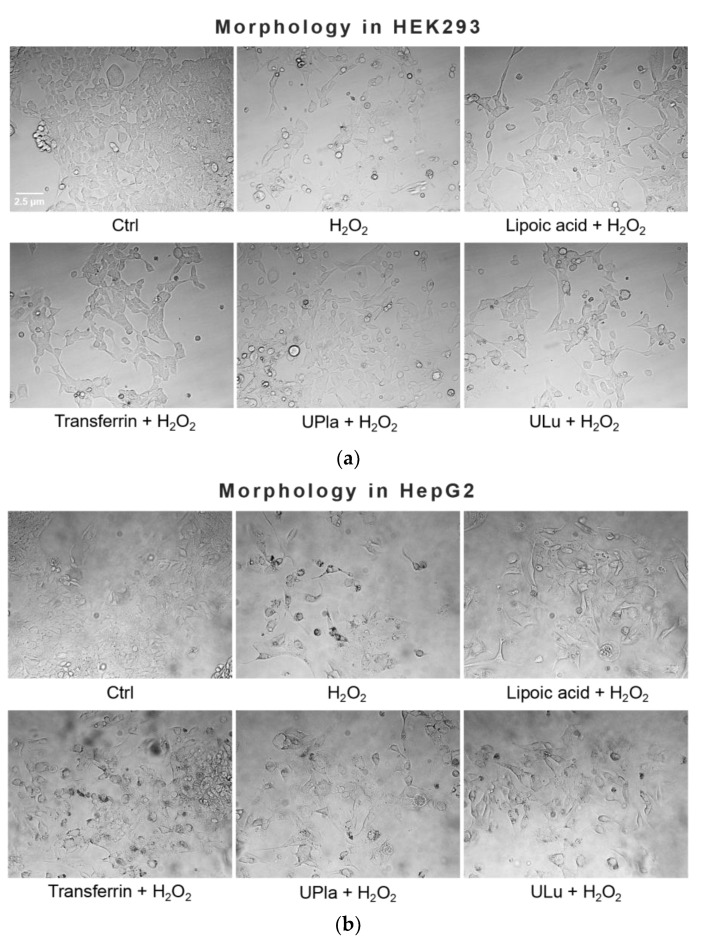
Effect of single substances on H_2_O_2_-induced cell morphology in HEK293 (**a**) and HepG2 cells (**b**). Normal cells were treated with H_2_O_2_ (300 μM) with or without single substances for three days, followed by four days of recovery culture. Afterwards, photos of the cells were taken at the magnification of 400× with a light microscope. The results represent three independent experiments.

**Figure 4 ijms-24-06748-f004:**
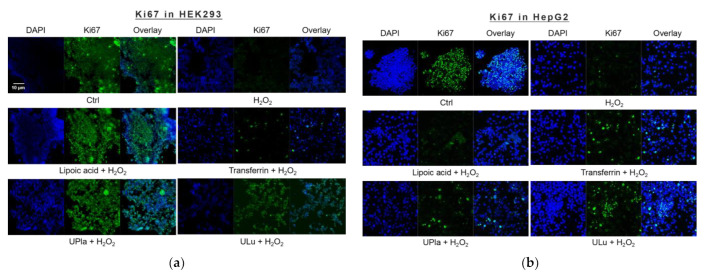
Cell proliferation changes in HEK293 (**a**) and HepG2 (**b**) cells upon H_2_O_2_ treatment and the addition of the substances. The blue color shows DAPI-stained cell nuclei; the green represents Ki67 antibody staining. (**a**) Cell proliferation was decreased due to H_2_O_2_ induction, which was reversed by treatment with UPla, Ulu, and the positive control lipoic acid in HEK293 cells; (**b**) UPla, Ulu, as well as transferrin reversed the cell proliferation decline in HepG2 cells. The results are representative of three independent experiments.

**Figure 5 ijms-24-06748-f005:**
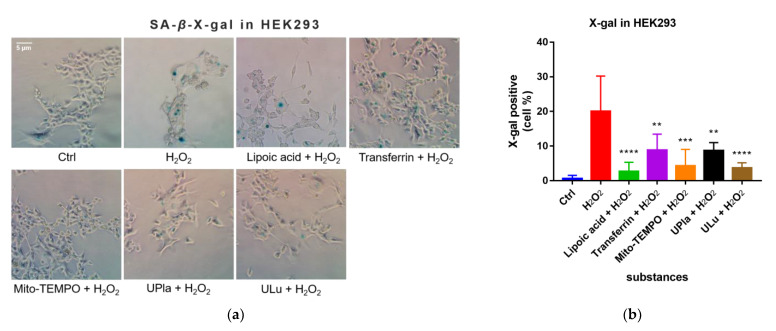
X-gal staining of the senescent HEK293 (**a**) and HepG2 cells (**c**) with or without single-substance treatment and the quantification of positively X-gal-stained HEK293 (**b**) and HepG2 (**d**) cells in comparison with the control groups. Cells were photographed at 400× magnification for HEK293 cells and 200× magnification for HepG2 cells. Data are presented as the mean ± SD, N = 3. * *p* < 0.05 vs. H_2_O_2_ treatment. ** *p* < 0.01, *** *p* < 0.001, **** *p* < 0.0001. Ctrl: control.

**Figure 6 ijms-24-06748-f006:**
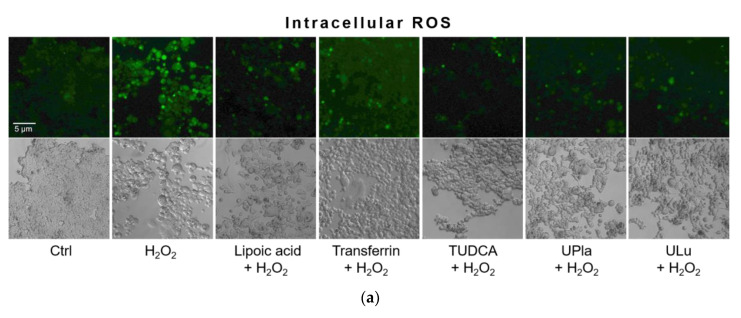
Intracellular ROS levels were detected by DCFH-DA staining (**a**), and the integrated fluorescence density was quantified using ImageJ (**b**). H_2_O_2_ raised intracellular ROS levels (green fluorescence), which were reduced by UPla or ULu. Similar effects were observed in the positive control groups with lipoic acid, transferrin, and TUDCA. The results represent similar results of three independent experiments. Values are the mean ± SD, N = 3. * *p* < 0.05 vs. H_2_O_2_ treatment, ** *p* < 0.01.

**Figure 7 ijms-24-06748-f007:**
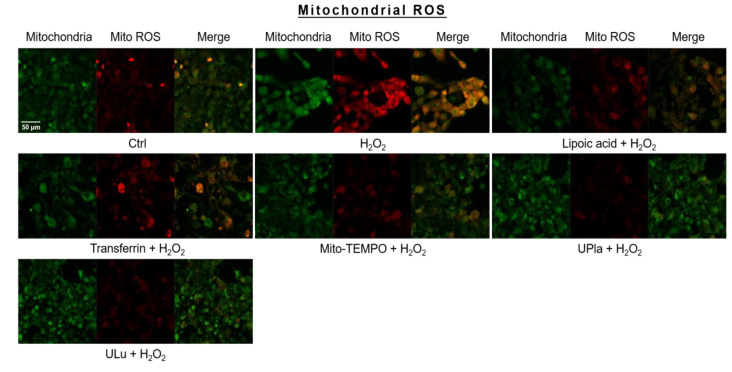
Mitochondrial ROS assessment with mitochondrial dye MitoTracker Green and mitochondrial ROS indicator MitoSox Red, with visualization by confocal microscopy. The results represent similar results of three independent experiments. Mito ROS: mitochondrial ROS.

**Figure 8 ijms-24-06748-f008:**
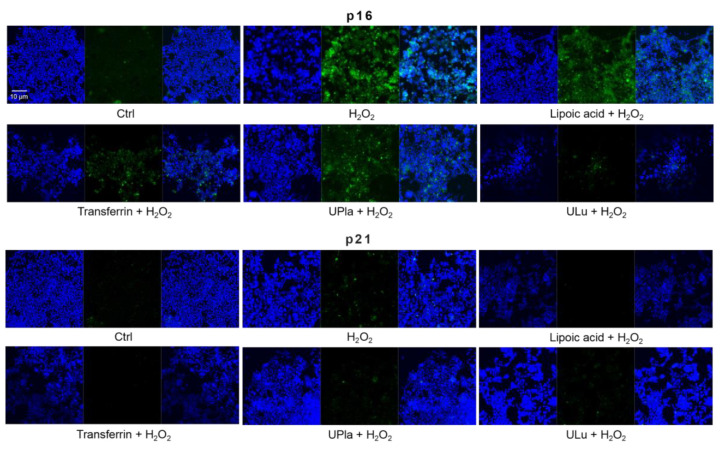
The expression of senescence markers was monitored by immunofluorescence staining in HEK293 cells upon treatment with H_2_O_2_, with or without single substances. Cells incubated with different senescence marker antibodies displayed green fluorescence and the nucleus was stained by DAPI. The results represent similar results from three independent experiments.

**Figure 9 ijms-24-06748-f009:**
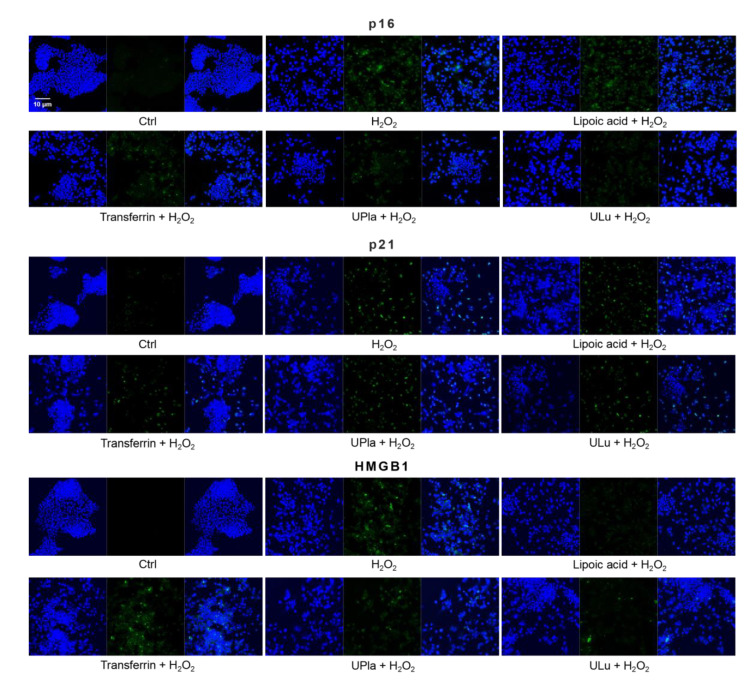
The expression of senescence markers was measured by immunofluorescence staining in HepG2 cells upon treatment with H_2_O_2_, with or without single substances. Cells were incubated with different senescence marker antibodies and counterstained by DAPI. The results represent similar results of three independent experiments.

**Figure 10 ijms-24-06748-f010:**
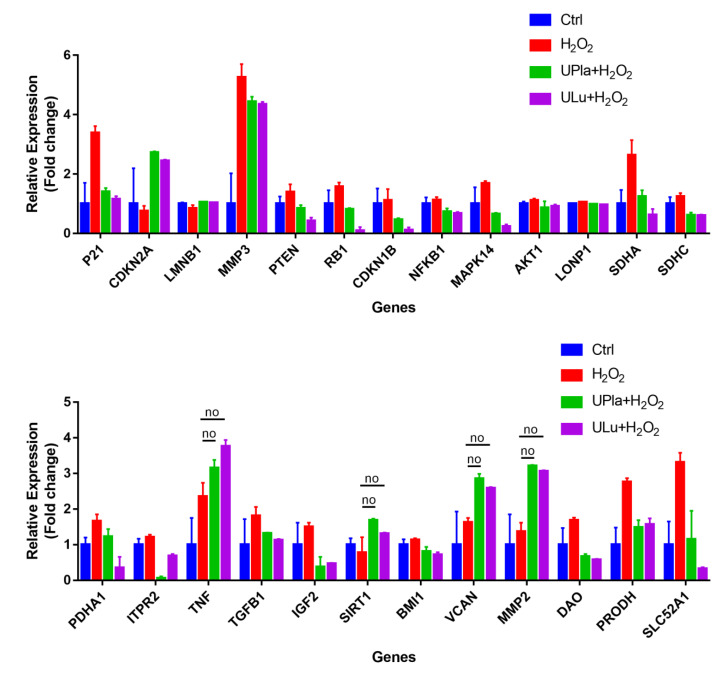
Gene expression profiling of H_2_O_2_-induced HepG2 senescence and modulation by UPla and ULu. Except genes labeled with “no”, the expression of all genes was significantly altered by H_2_O_2_ and significantly modulated by Upla and Ulu. The results are shown as fold changes to the control groups. The results are expressed as the mean ± SD from three sets of independent samples. Significance was indicated when *p* < 0.05. No: UPla or ULu did not affect gene expression changes.

**Figure 11 ijms-24-06748-f011:**
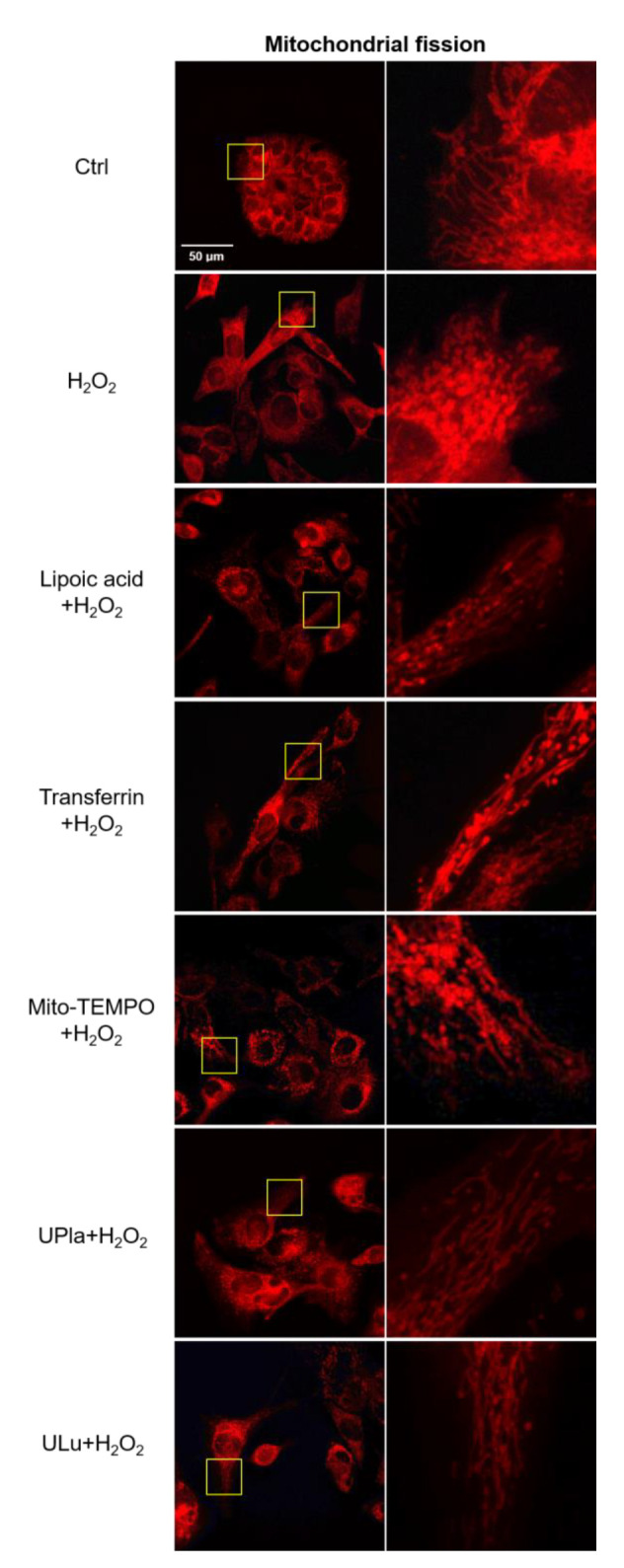
Mitochondrial fission was detected with the mitochondrial dye MitoTracker Red and viewed by confocal microscopy. On the right side, the areas in the rectangle (left column) are magnified. Spherical or fragmented mitochondria were regarded as undergoing fission. The images are representative of three independent experiments.

**Table 1 ijms-24-06748-t001:** TaqMan gene assays.

#	Gene	Catalog Number Thermo Fisher	Task
1	18S	Hs99999901_s1	Reference gene
2	HMBS	Hs00609297_m1	Reference gene
3	GAPDH	Hs99999905_m1	Reference gene
4	TP53	Hs01034249_m1	Tumor suppressor gene
5	CDKN1A/P21	Hs01040810_m1	Cyclin-dependent kinase inhibitor 1A
6	CDKN2A/p16	Hs00923894_m1	Cyclin-dependent kinase inhibitor 2A
7	HMGB1	Hs07287366_m1	Senescence marker
8	LMNB1	Hs01059205_m1	Senescence marker
9	MMP3	Hs00968305_m1	Senescence marker, SASP
10	CHEK2	Hs00200485_m1	Tumor suppressor gene
11	PTEN	Hs02621230_s1	Tumor suppressor gene
12	E2F1	Hs00153451_m1	Transcription factor, tumor suppressor gene, cell cycle
13	RB1	Hs01078066_m1	Tumor suppressor gene, signaling pathway
14	GADD45A	Hs00169255_m1	Cell cycle arrest
15	CDKN1B	Hs00153277_m1	Cyclin-dependent kinase inhibitor 1B (p27^Kip1^)
16	CDK4	Hs00364847_m1	Cyclin-dependent kinase 4
17	CDC25C	Hs00156411_m1	M-phase inducer phosphatase 3
18	HRAS	Hs00610483_m1	GTPase Hras, cell division, signaling
19	NFKB1	Hs00765730_m1	Signaling pathway
20	MAPK14	Hs01051152_m1	Signaling pathway
21	AKT1	Hs00178289_m1	Signaling pathway
22	LONP1	Hs00998404_m1	Mitochondrial protein
23	SDHA	Hs07291714_mH	Mitochondrial protein
24	SDHB	Hs01042482_m1	Mitochondrial protein
25	SDHC	Hs00818427_m1	Mitochondrial protein
26	PDHA1	Hs01049345_g1	Mitochondrial protein
27	ITPR2	Hs00181916_m1	Calcium channel
28	IL6	Hs00174131_m1	Secreted cytokines, SASP
29	TNF	Hs00174128_m1	Secreted cytokines, SASP
30	TGFB1	Hs00998133_m1	Transforming growth factor beta 1, SASP
31	IGF2	Hs04188276_m1	Insulin-like growth factor
32	SIRT1	Hs01009006_m1	Longevity protein
33	BMI1	Hs00409821_g1	Polycomb ring finger oncogene
34	VCAN/CSPG2	Hs00171642_m1	[41]
35	COL1A1	Hs00164004_m1	[41]
36	MMP2	Hs01548727_m1	[41], SASP
37	DAO	Hs00266481_m1	[41]
38	PRODH	Hs01574361_g1	[42]
39	SLC52A1/GPR172B	Hs00606016_g1	[42]
40	DLP1	Hs01552605_m1	Mitochondrial fission

**Table 2 ijms-24-06748-t002:** Genes significantly upregulated by H_2_O_2_ in HepG2 senescence.

Gene	Task	Fold Change	UPla	ULu
CDKN1A/P21	Cyclin-dependent kinase inhibitor 1A	3.39	Yes *	Yes
MMP3	Senescence marker, SASP	5.26	Yes	Yes
PTEN	Tumor suppressor gene	1.39	Yes	Yes
RB1	Tumor suppressor gene, signaling pathway	1.57	Yes	Yes
CDKN1B/p27	Cyclin-dependent kinase inhibitor 1B	1.11	Yes	Yes
NFκB1	Signaling pathway	1.13	Yes	Yes
MAPK14	Signaling pathway	1.68	Yes	Yes
AKT1	Signaling pathway	1.11	Yes	Yes
LONP1	Mitochondrial protein	1.05	Yes	Yes
SDHA	Mitochondrial protein	2.64	Yes	Yes
SDHC	Mitochondrial protein	1.24	Yes	Yes
PDHA1	Mitochondrial protein	1.66	Yes	Yes
ITPR2	Calcium channel	1.21	Yes	Yes
TNF	Secreted cytokines, SASP	2.35	No	No
TGFB1	Transforming growth factor beta 1, SASP	1.81	Yes	Yes
IGF2	Insulin-like growth factor	1.50	Yes	Yes
BMI1	Polycomb ring finger oncogene	1.14	Yes	Yes
VCAN/CSPG2	Induction greater than 4-fold in HPEC senescence [41]	1.63	No	No
MMP2	SASP, induction greater than 4-fold in HPEC senescence [41]	1.37	No	No
DAO	Induction greater than 4-fold in HPEC senescence [41]	1.69	Yes	Yes
PRODH	Identified in senescence depending on p53 [42]	2.77	Yes	Yes
SLC52A1/GPR172B	Identified in senescence depending on p53 [42]	3.32	Yes	Yes

Gene expression was normalized to HMBS expression. Fold change is the ratio of expression in senescent cells to the expression in the control groups. * Yes: Upla or Ulu modulated H_2_O_2_-caused gene expression alterations. No: UPla or ULu did not affect gene expression changes.

**Table 3 ijms-24-06748-t003:** Genes significantly downregulated by H_2_O_2_ in HepG2 senescence.

Gene	Task	Fold Change	UPla	ULu
CDKN2A/p16	Cyclin-dependent kinase inhibitor 2A	0.75	Yes	Yes
LMNB1	Senescence marker	0.84	Yes	Yes
SIRT1	Longevity protein	0.78	Yes	No

## Data Availability

Data is contained within the article or Appendix A.

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
