# Peer review of "Modulation of Cellular Senescence in HEK293 and HepG2 Cells by Ultrafiltrates UPla and ULu Is Partly Mediated by Modulation of Mitochondrial Homeostasis under Oxidative Stress"

_ijms, 2023, doi:10.3390/ijms24076748_

Round 1
Reviewer 1 Report
The topic of the paper „Modulation of cellular senescence in HEK293 and HepG2 cells by ultrafiltrates UPla and ULu is partly mediated by modulation of mitochondrial homeostasis under oxidative stress” is very interesting for readers, protein probes, -ultrafiltrates from the placenta and ultrafiltrates from the lung of post-natal rabbits-, being investigated in premature senescent HEK293 and HepG2 cells to explore whether they could modulate cellular senescence.
The authors show that the obtained results demonstrate that ultrafiltrates from the placenta and ultrafiltrates from the lung, as well as lipoic acid and transferrin, could protect HEK293 and HepG2 cells from H2O2 induced oxidative damage via protecting mitochondrial homeostasis and have the potential to be explored in anti-aging therapy.
The introduction provides sufficient background and includes relevant references.
The manuscript is well written, and the text is easy to read.
The design research is well described.
The results are consistent and clearly presented.
The reference list is variously and relatively recently.
Reviewer 2 Report
This study investigated the role of ultrafiltrates UPla and Ulu in cellular senescence. It demonstrated that the UPla and Ulu protect cells from oxidative stress by protecting mitochondrial homeostasis. The article is well organized, and its presentation is good. However, some technical issues still need to be improved before this can be considered for publication.
The cell morphology results in Figure 3 need to be improved with the magnification scale bar on the figure.
All the microscopy figures (figure 4; 5a, c; 6a; 7; 8, 9, and 11) are missing the scale bar.
Figure 10, gene expression p-value needs to be included on the graph.
Multiple typos need to be corrected. Include full gene labels on figures like Lamin B1.
The methods, 4.7 Bioinformatics text needs to be formatted.
Reviewer 3 Report
The authors demonstrate protective effects of UPla and Ulu against H2O2 induced oxidative stress and cell senescence. The experiments were designed clearly, and results are complete and solid. The background introduction is not very sufficient, some background information about UPla and Ulu clinical application and why authors focused on these two protein probes should be included.
Round 2
Reviewer 2 Report
The authors have improved the manuscript.